# The Effects of Ying Yang Bao on Nutritional Status of Children Aged 6–60 Months in Underdeveloped Rural Areas of China

**DOI:** 10.3390/nu16020202

**Published:** 2024-01-08

**Authors:** Jing Feng, Yongjun Wang, Tingting Liu, Junsheng Huo, Qin Zhuo, Zhaolong Gong

**Affiliations:** 1Key Laboratory of Trace Element Nutrition of National Health Commission, National Institute for Nutrition and Health, Chinese Center for Disease Control and Prevention, Beijing 100050, China; fengjing0921@163.com (J.F.); wangyongjun519@163.com (Y.W.); liutt@ninh.chinacdc.cn (T.L.); huojs@ninh.chinacdc.cn (J.H.); 2Department of Clinical Nutrition, The First Affiliated Hospital of Shandong First Medical University & Shandong Provincial Qianfoshan Hospital, Jinan 250014, China

**Keywords:** Ying Yang Bao, undernutrition, anemia, two-level regression model, long-term effects

## Abstract

The Ying Yang Bao (YYB) intervention, a national policy in China, has been implemented for over two decades. Most previous studies have focused only on the short-term effects of YYB, while the long-term effects remain unexplored. This study was designed to evaluate the long-term effects of YYB in children aged 6–60 months. A sample of 4666 children was divided into intervention and control groups. Information on basic characteristics, physical examination, YYB consumption, etc., was obtained annually from 2018 to 2021. T-tests or chi-square tests were used to compare differences between the groups for continuous or categorical variables. Children in the intervention group showed greater incremental improvements in hemoglobin levels and physical development (*p* < 0.05). Prevalence of anemia, underweight, and stunting were lower in the intervention group than in the control group (all *p* < 0.05). Two-level regression models were constructed to assess the long-term effects of YYB. YYB reduced the risk of anemia and wasting by 37% (OR: 0.63, 95% CI: 0.52–0.75) and 49% (OR: 0.51, 95% CI: 0.39–0.67), respectively. This study indicates that YYB could significantly improve the nutritional status of children aged 6–60 months in underdeveloped rural areas of China.

## 1. Introduction

Undernutrition is a major health problem in early childhood. Undernutrition includes four broad sub-forms: wasting, stunting, underweight, and deficiencies in vitamins and minerals [1,2,3]. Globally in 2020, 149 million children under five were estimated to be stunted, and 45 million were wasted [3]. At the same time, anemia remains a leading cause of childhood morbidity and mortality in low- and middle-income countries and regions, affecting an estimated 280 million children (approximately 40%) under five globally [4]. According to the Report on Nutrition and Chronic Diseases in China (2020), the prevalence of stunting, underweight, wasting, and anemia among children under six in Chinese rural areas were 18.7%, 5.2%, 3.0%, and 25.6%, respectively, in contrast to the prevalence of 3.5%, 1.5%, 1.7%, and 15.0% in urban areas [5]. In addition, infants and young children (IYC) aged 6–24 months have a higher anemia rate than other age groups. The prevalence of anemia in 6- to 24-month-old IYC of China was 36.9% overall and up to 42.0% in rural areas [5,6].

Undernutrition in children aged 6–60 months may have irreversible effects on their development [7,8,9], including more frequent and severe infections [10,11], impaired motor and cognitive development [12,13], and decreased academic performance [14]. The nutritional status of Chinese children has gradually improved over the past decade. However, significant gaps remain between urban and rural areas [5]. Improving nutritional status of children in underdeveloped rural areas has always been the top priority in China’s nutrition improvement work. To improve the health status of children, many countries and regions have implemented intervention measures, such as the Special Supplemental Nutrition Program for Women, Infants and Children in the United States [15] and the Integrated Child Development Services in India [16]. In China, to improve the nutritional and health status among IYC aged 6–24 months, the Nutrition Improvement Project on Children in Poor Areas of China (NIPCPAC) was launched in 2012 by the Ministry of Health and the All-China Women’s Federation. The NIPCPAC provides a free bag of nutrition pack (called Ying Yang Bao in Chinese, YYB) daily for IYC aged 6–24 months in underdeveloped rural areas of China [17,18,19].

The YYB intervention, a national policy in China, has been described in our previous articles [6,20]. Briefly, YYB is a home-based complementary food fortification that contains a variety of nutrients, such as proteins, calcium, iron, zinc, vitamin A, vitamin C, vitamin D, folic acid, etc. And, the duration of the YYB intervention for IYC aged 6–24 months is 18 months. Many studies have demonstrated the improvement effect of YYB on the nutritional status of IYC aged 6–24 months. YYB was effective in reducing the prevalence of anemia, underweight, and other malnutrition-related developmental delays [21,22,23]. To continuously monitor the nutritional status of children aged ≥ 24 months, the Institute of Nutrition and Health of the Chinese Center for Disease Control and Prevention launched the Long-term Health Effects Assessment Project of Infants and Toddlers Nutrition Pack (LHEAPITNP) in 2018. This project is a prospective controlled trial that aims to explore the short- and long-term effects of YYB on children’s nutritional status. The LHEAPITNP focuses mainly on the nutritional status of children, including their hemoglobin (Hb) and anemia levels, physical and behavioral development, etc.

Previous studies on YYB have mostly been self-controlled before and after intervention, without parallel no-treatment control groups. Moreover, most studies have focused only on the nutritional status of IYC who were taking YYB, and on the short-term health effects of YYB. There have been few reports on children who finished taking YYB, or on the long-term effects of YYB.

Based on multi-regional data from LHEAPITNP (2018–2021), this study set a parallel no-treatment control group, controlled for confounding factors, and discussed the long-term effects of YYB among children aged 6–60 months in underdeveloped rural areas of China.

## 2. Materials and Methods

### 2.1. Study Design and Population

Sample selection and data collection procedures were as reported elsewhere [6,19]. Briefly, the LHEAPITNP sampled 6- to 24-month-old IYC in the surveillance areas of the NIPCPAC through multi-stage sampling in 2018 [19]. The LHEAPITNP sampled 10 YYB intervention-covered counties as the intervention groups: Guiding, Songxian, Moyu, Qichun, Longhua, Tongyu, Yunlong, Huangzhong, Yanchi, and Shilou. Simultaneously, 5 counties with similar geographical locations and economic conditions that were not covered by the YYB intervention were sampled as controls: Fuquan, Ruyang, Luopu, Huangmei, and Weichang. Approximately 300 IYC were randomly selected from each sample county.

The following children were excluded from recruitment to the LHEAPITNP: a. migrant children; b. children suffering from infectious diseases, severe acute malnutrition, known stunting, or any other chronic diseases; c. children with anaphylaxis to soy protein or milk; and d. children in a YYB intervention-covered area but not taking YYB. The detailed sampling procedure was described in our previous studies [6,19].

Follow-up surveys were conducted from 2018 onwards, including questionnaire investigations, physical examinations, and biological sample collection. Information on the basic characteristics of children and parents, the nutritional knowledge level of caregivers, the average dietary diversity of children in the last 24 h, YYB consumption, Hb levels, and measurements of children’s physique were collected by staff working at the maternity and child healthcare hospitals.

This study further excluded subjects without information on YYB consumption or on basic characteristics. Those beyond 6–24 months at recruitment were also excluded. Finally, 4666 subjects were included in the follow-up analyses. An overview of the inclusion and exclusion processes is presented in Figure 1.

### 2.2. Ethical Issues

The LHEAPITNP was reviewed and approved by the Ethics Committee of the Institute of Nutrition and Health of the Chinese Center for Disease Control and Prevention (No. 2018-017). The caregivers of all participating children were fully informed, and signed informed consent forms.

### 2.3. Indicators and Definitions

#### 2.3.1. Outcomes

According to the World Health Organization (WHO), the prevalence of anemia was defined as altitude-adjusted Hb levels <110 g/L among children aged under 5 years (6–59.9 months old) and <115 g/L among children aged 5 years (60–71.9 months old) [24]. Physical development indicators included: weight, length or height, weight-for-age z-score (WAZ), length/height-for-age z-score (LAZ/HAZ), and weight-for-length/height z-score (WLZ/WHZ). Undernutrition included underweight, stunting, and wasting [25]. According to the WHO Child Growth Standards [26,27], underweight was defined as WAZ of <−2 standard deviations (SDs) of the median; stunting was defined as LAZ < −2 SDs; wasting was defined as WLZ < −2 SDs (for children aged under 5 years) or BMI Z score < −2 SDs (for children aged 5 years).

#### 2.3.2. Covariates

Based on the LHEAPITNP design and previous studies [6,28], the following were included as covariates in this study: sex (male or female), age, birth weight, birth length, premature birth (yes or not), nutritional status at 6 months, average dietary diversity, consumption of other supplements (yes or not), main caregiver (parents or others), caregiver’s education (primary school or below, middle school, high school or above), caregiver’s knowledge of nutritional feeding, family economic status, regional health services, and regional economic situation.

The baseline status of children in the current study was defined as the nutritional status at 6 months of age, including Hb levels, weight, length, WAZ, LAZ, WLZ, anemia (yes or not), underweight (yes or not), stunting (yes or not), and wasting (yes or not). Almost half of the respondents chose the responses of “refuse to tell” or “do not know” for the questions on annual income and expenditure of their family because of privacy. The current study referred to previous studies and calculated the socio-economic status (SES) according to the parental educational and occupation [29,30].

### 2.4. Statistical Analysis

Data were reported as mean ± SD for continuous variables and as number (frequencies) for categorical variables. *T*-tests and chi-square tests were used to evaluate the difference of basic characteristics between the intervention group and the control group. Additionally, *t*-tests were used to compare differences in Hb levels, weight, length/height, and Z-scores between the intervention and control groups. Chi-square tests were used to compare the prevalence of anemia, underweight, stunting, and wasting.

Additionally, to access the long-term effects (i.e., repeated measurements clustered within participants), differences in nutritional status between the intervention and control groups were assessed using a two-level regression model. A two-level regression model is one of the multi-level models (MLMs). MLM is a class of models for processing hierarchical data. It provides proper parameter estimates and standard errors for clustered data [31]. In this study, two-level linear regression models and two-level logistic regression models were constructed for continuous and categorical variables, respectively, with the time level as level 1 and the individual level as level 2. Two models were established for each outcome measure. Models I included only time and grouping, and rough analysis was performed; Models II included covariates that were significant across groups to obtain the adjusted parameters.

SPSS (version 19.0; IBM Institute, Albany, NY, USA) was used for conducting t-tests and chi-square tests. The “lme4 package” in R software (version 4.1.2) was used to construct two-level regression models [32]. Prism software package (version 8.3; GraphPad Software Inc., San Diego, CA, USA) was used for plotting. Differences were considered statistically significant at *p* < 0.05.

## 3. Results

### 3.1. Basic Characteristics of Subjects

As shown in Table 1, 4666 subjects were divided into the intervention (*n* = 3034) and the control groups (*n* = 1632) based on whether they received YYB. The average age at recruitment was 14.3 months, and 52.0% were male. Average birth weight and length were 3278.1 g and 50.1 cm, respectively, and 4.1% were born prematurely. Of these children, 83.2% of children were primarily cared for by their parents. The average score for the caregiver’s knowledge of nutritional feeding was 2.8 (out of 5). Children consumed an average of 5.1 foods per day. In addition, 15.1% of the subjects consumed other supplements other than YYB.

There were no significant differences between the control and intervention groups in age, sex, birth weight, birth length, and premature birth (all *p* > 0.05). Children in the intervention group had a lower SES (50.4 vs. 50.9, *p* = 0.02) and a higher proportion of parental care (84.0% vs. 81.7%, *p* = 0.04). More caregivers had a high school education or above in the intervention group (25.6% vs. 22.3%, *p* = 0.01). Compared with the children in the control group, those in the intervention group consumed more diversified foods (5.2 vs. 4.9, *p* < 0.01), and their caregivers scored higher on knowledge of nutritional feeding (3.0 vs. 2.6, *p* < 0.01). Children in the intervention group were less likely to consume other supplements (11.5% vs. 21.9%, *p* < 0.05).

### 3.2. Nutritional Status of Children Aged 6–60 Months

Generally, the nutritional status of children in the intervention group was significantly better than that in the control group (Table 2). Children in the intervention group had higher average Hb levels (124.7 vs. 123.1), weight (13.0 vs. 12.7), length/height (90.7 vs. 89.8), WAZ (−0.1 vs. −0.2), LAZ/HAZ (0.0 vs. −0.2), and WLZ/WHZ (−0.1 vs. −0.2) than those in the control group (all *p* < 0.01). Additionally, the prevalence of anemia, underweight, and stunting of children in the intervention group were 11.1%, 2.5%, and 3.2%, respectively, significantly lower than the 12.4%, 3.0%, and 5.4% in the control group (*p* values were 0.02, 0.04 and <0.01, respectively). There was no significant difference in wasting between the two groups (*p* = 0.54).

Figure 2 shows the mean change in the nutritional status of children at each annual visit by group. As shown in Figure 2A,B, the prevalence of anemia in the intervention group was significantly lower than that in the control group in 2018 and 2019 (*p* < 0.05), and was essentially the same as that in the control group in 2020 and 2021.

From 2018 to 2021, children in the control group showed significantly greater reductions in WAZ and WLZ/WHZ than those in the intervention group (Figure 2E,G). The prevalence of stunting decreased and wasting increased in the control group (Figure 2H,J). The prevalence of undernutrition among children in the intervention group remained low in each year (Figure 2H–J). In both groups, the prevalence of underweight and wasting was <5% from 2018 to 2021 (Figure 2H,J).

### 3.3. Two-Level Regression Analysis of the Effects of YYB

The results of Models I and Models II were consistent, as shown in Table 3. In Models II, after adjusting for confounders, the differences in nutritional status among individuals were lower than those in Models I. Results of the adjusted models showed that YYB increased the Hb levels (*β*: 1.36, 95% CI: 0.92–1.81) and reduced the anemia rate (OR: 0.63, 95% CI: 0.52–0.75).

After adjusting for confounders, YYB effectively promoted WAZ (*β*: 0.16, 95% CI: 0.10–0.21), LAZ/HAZ (*β*: 0.18, 95% CI: 0.12–0.24), and WLZ/WHZ (*β*: 0.09, 95% CI: 0.03–0.14) scores. In addition, YYB reduced the risk of stunting (OR: 0.51, 95% CI: 0.39–0.67) in children aged 6–60 months, but a similar effect was not found on underweight (OR: 0.77, 95% CI: 0.56–1.06) and wasting (OR: 1.04, 95% CI: 0.80–1.37).

## 4. Discussion

Children aged 6–60 months in the intervention group showed greater incremental improvements in Hb levels, weight, length/height, and Z-scores, as well as lower rates of anemia, underweight, and stunting than children in the control group. From the results of the covariation-adjusted two-level regression models, the YYB intervention reduced the risk of anemia and stunting in children aged 6–60 months by 37% and 49%, respectively. No correlation was observed between YYB intervention and underweight and wasting in this study. Our results suggest that the 18-month YYB intervention has a long-term and far-reaching impact on reducing the risk of anemia and undernutrition, and promoting physical development in children aged 6–60 months.

In this study, for the first time, the long-term health effects of YYB were evaluated. YYB, like other multi-nutrient supplements, is a low-cost measure recognized by the WHO as effective in improving the nutritional status of children [33]. Multiple studies have proved that YYB has a positive short-term effect in improving the nutritional status of IYC. Yin et al. [34] provided iron-fortified soy sauce, dietary supplements, and YYB to pregnant women, lactating women, and 6-month-old infants in a 2.4-year intervention. Their results showed that comprehensive intervention measures could reduce the anemia rate of infants by 40%. Studies in other countries and regions have also affirmed the role of nutritional supplements in improving children’s nutritional status. A nutritional intervention trial in 24- to 59-month-old Iranian children found that supplemental feeding of ready-to-use supplementary foods for 8 weeks significantly improved weight, height, and BMI, and reduced the prevalence of diarrhea and fever [35]. Further, the results from a randomized controlled trial conducted by Roberts et al. in rural Guinea showed that a 23-week intervention with nutritional supplements improved the cognitive function, brain health, and nutritional status of vulnerable infants [36].

The positive effects of YYB may be partly due to the variety of nutrients it contains. YYB is rich in minerals, vitamins, proteins, and other nutrients [20]. The high-quality proteins in YYB are derived from soybeans or milk and are essential for body function. Moreover, micronutrients can also enable the body to produce enzymes, hormones, and other substances that are essential for proper growth and development [3]. Among these various nutrients in YYB, iron and folic acid are important in preventing and treating nutritional anemia. And, vitamin C and vitamin D3 can promote the absorption of minerals such as iron, zinc, and calcium [37]. The various nutrients in YYB are effective to prevent malnutrition and promote child development. Therefore, YYB is a necessary supplement for children in less-developed rural areas of China, where the rate of acceptable complementary feeding is low [18].

Higher health literacy among caregivers, promoted by the NIPCPAC policy, may also partly explain the positive effects observed in this study. In addition to the direct health effects of nutrients in YYB, the promotion of NIPCPAC policies has led to a greater focus on the children’s growth process by caregivers, and indirectly improved complementary feeding skills among caregivers [38,39]. Research conducted by Wang et al. found that caregivers in the intervention group achieved higher scores for nutritional feeding, and children in the intervention group were more likely to meet the requirements of minimum dietary diversity, minimum meal frequency, and minimum acceptable diet [40]. Yao et al. [39] carried out child health education through home visits in Liangshan, Sichuan Province. Their staff simultaneously conducted child health consultations and YYB interventions. Their intervention significantly improved child compliance in taking YYB, increased the acceptance rate of complementary foods, and increased the proportion of regular physical examinations and vaccinations [39].

Our study observed no statistical significance between the YYB intervention and underweight and wasting, which was consistent with previous studies by Xu [41] and Wang et al. [23]. On the one hand, weight gain depends mainly on the balance of energy intake and expenditure [42]. Many other factors—including dietary behavior, infectious diseases, family circumstances—also contribute to weight changes in children [43,44]. For example, Chen et al. [45] indicated that insufficient milk intake was one of the major reasons for poor weight gain in Chinese infants. On the other hand, many reports based on national surveillance data have shown that the wasting rate among Chinese children under the age of five has remained low for a long time [5,46,47]. In addition, Xu argued that wasting was a serious condition that reflected acute malnutrition in children, and that sudden illness may have a greater impact on it [41]. The YYB intervention may thus be limited in reducing the prevalence of wasting.

Our study has several strengths. First, this study evaluated the long-term effects of YYB in children aged 6–60 months, which have not been previously studied. Previous studies have focused only on the short-term effects of YYB on IYC aged 6–24 months. Second, this study population was representative of the children in Chinese less-developed rural areas. This study was a prospective, controlled study with a sample of children from the eastern, central, western, and northeastern regions of China.

This study also has its limitations. Firstly, the areas covered by the NIPCPAC are publicly known, and the distribution of YYB can also be determined. Blindness was not used in the present study. In addition, to allow for comparability between the intervention and control groups, the two groups in the same province were adjacent counties, which may have interfered with each other.

## 5. Conclusions

This study suggests that YYB has a positive and lasting effect on improving the nutritional status of children aged 6–60 months. The 18-month YYB intervention effectively promoted physical development and reduced the risk of anemia and stunting for children aged 6–60 months. The NIPCPAC provides nutritional and technical support to children in less-developed areas. This study provides scientific data for the implementation of Chinese nutritional intervention policies.

## Figures and Tables

**Figure 1 nutrients-16-00202-f001:**
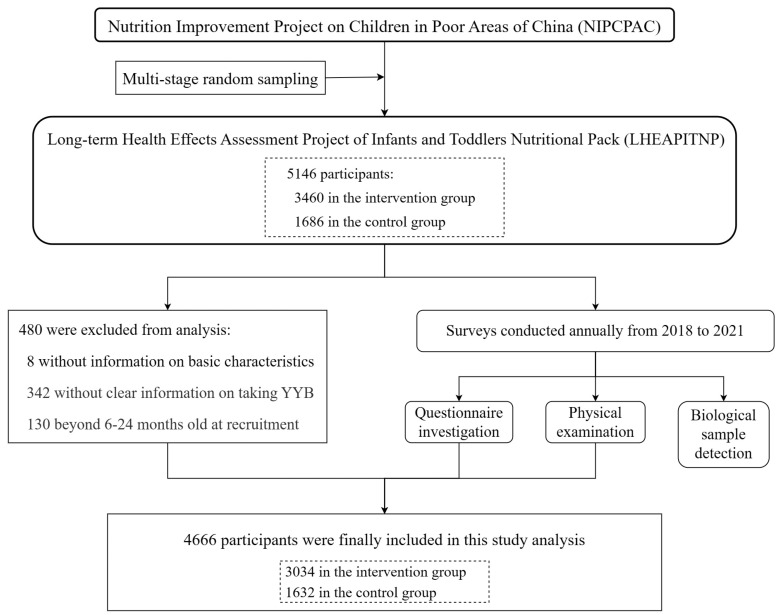
Procedures for subject recruitment and data collection.

**Figure 2 nutrients-16-00202-f002:**
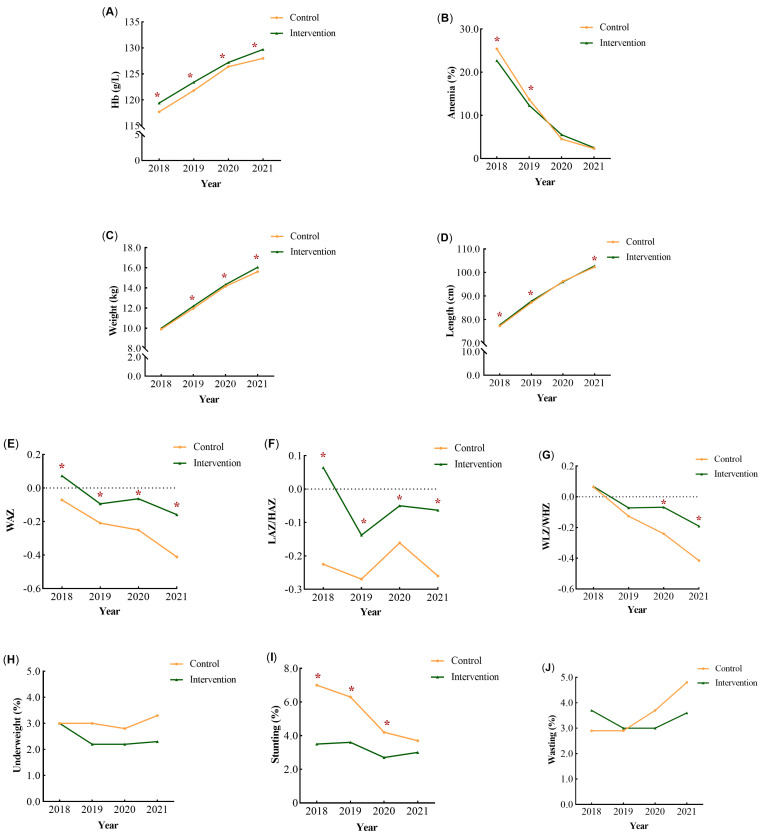
The mean change in the nutritional status of children from 2018 to 2021. The mean change in the Hb levels (**A**), anemia rate (**B**), weight (**C**), length or height (**D**), WAZ (**E**), LAZ/HAZ (**F**), WLZ/WHZ (**G**), underweight rate (**H**), stunting rate (**I**), and wasting rate (**J**) were shown at each annual visit by group. * *p* < 0.05.

**Table 1 nutrients-16-00202-t001:** Basic characteristics of subjects.

Characteristics	All Subjects (*n* = 4666)	Control (*n* = 1632)	Intervention (*n* = 3034)	*p* Value
Age at recruitment (months)	14.3 ± 5.3	14.5 ± 5.4	14.2 ± 5.2	0.07
Sex, *n* (%)				0.71
Male	2424 (52.0)	854 (52.3)	1570 (51.7)	
Female	2242 (48.0)	778 (47.7)	1464 (48.3)	
Birth weight (g)	3278.1 ± 505.8	3292.4 ± 509.6	3270.5 ± 503.7	0.16
Birth length (cm)	50.1 ± 1.7	50.1 ± 1.8	50.1 ± 1.6	0.16
Premature birth, *n* (%)	192 (4.1)	69 (4.2)	123 (4.1)	0.78
SES	50.6 ± 7.4	50.9 ± 7.0	50.4 ± 7.6	0.02
Parents as main caregivers, *n* (%)	3883 (83.2)	1333 (81.7)	2550 (84.0)	0.04
Caregiver’s education, *n* (%)				0.01
Primary school or below	960 (21.0)	359 (23.0)	601 (20.0)	
Middle school	2489 (54.5)	857 (54.8)	1632 (54.4)	
High school or above	1116 (24.4)	348 (22.3)	768 (25.6)	
Caregiver’s knowledge of nutritional feeding	2.8 ± 1.1	2.6 ± 1.0	3.0 ± 1.1	<0.01
Average dietary diversity	5.1 ± 1.0	4.9 ± 1.1	5.2 ± 1.0	<0.01
Consume other supplements, *n* (%)	706 (15.1)	357 (21.9)	349 (11.5)	<0.01

SES, socio-economic status.

**Table 2 nutrients-16-00202-t002:** Nutritional status of children aged 6–60 months in the two groups.

Outcome Measures	All Subjects	Control	Intervention	*p* Value
Hb levels and anemia				
Hb levels (g/L)	124.2 ± 11.1	123.1 ± 10.9	124.7 ± 11.2	<0.01
Anemia, *n* (%)	1852 (11.5)	649 (12.4)	1203 (11.1)	0.02
Physical development				
Weight (kg)	12.9 ± 3.0	12.7 ± 2.9	13.0 ± 3.0	<0.01
Length/Height (cm)	90.4 ± 11.0	89.8 ± 11.1	90.7 ± 10.9	<0.01
WAZ	−0.1 ± 1.0	−0.2 ± 1.0	−0.1 ± 1.0	<0.01
LAZ/HAZ	−0.1 ± 1.1	−0.2 ± 1.2	0.0 ± 1.1	<0.01
WLZ/WHZ	−0.1 ± 1.1	−0.2 ± 1.0	−0.1 ± 1.1	<0.01
Undernutrition				
Underweight, *n* (%)	424 (2.6)	158 (3.0)	266 (2.5)	0.04
Stunting, *n* (%)	629 (3.9)	284 (5.4)	345 (3.2)	<0.01
Wasting, *n* (%)	541 (3.4)	183 (3.5)	358 (3.3)	0.54

Hb, hemoglobin; WAZ, weight-for-age z-score; LAZ/HAZ, length/height-for-age z-score; WLZ/WHZ, weight-for-length/height z-score.

**Table 3 nutrients-16-00202-t003:** Effects of Ying Yang Bao on the nutritional status of children.

Outcome Measures	Models I		Models II	
	β/OR (95% CI)	ICC (%)	β/OR (95% CI)	ICC (%)
Hb levels	1.30 (0.85, 1.75) *	88.3	1.36(0.92, 1.81) *	87.3
Anemia	0.64 (0.53, 0.76) *	61.1	0.63(0.52, 0.75) *	60.5
Weight	0.21 (0.10, 0.31) *	39.5	0.24(0.15, 0.33) *	30.5
Length/Height	0.28 (−0.03, 0.59)	86.7	0.47(0.26, 0.68) *	65.2
WAZ	0.16 (0.11, 0.22) *	15.8	0.16(0.10, 0.21) *	15.7
LAZ/HAZ	0.17 (0.11, 0.23) *	17.4	0.18(0.12, 0.24) *	16.9
WLZ/WHZ	0.10 (0.05, 0.16) *	14.2	0.09(0.03, 0.14) *	12.8
Underweight	0.74 (0.54, 1.01)	73.7	0.77(0.56, 1.06)	74.3
Stunting	0.54 (0.42, 0.70) *	71.4	0.51(0.39, 0.67) *	71.9
Wasting	1.02 (0.78, 1.32)	69.1	1.04(0.80, 1.37)	69.7

ICC, intraclass correlation coefficient; OR, odds ratio; CI, confidence interval; WAZ, weight-for-age z-score; LAZ/HAZ, length/height for age z-score; WLZ/WHZ, weight-for-length/height z-score. Models I included only time and grouping. Models II adjusted for covariates that were significant across groups, including age at recruitment, SES, caregiver, caregiver’s education, caregiver’s knowledge of nutritional feeding average dietary diversity, consumption of other supplements. * *p* < 0.05.

## Data Availability

The datasets generated and analyzed during the current study are not publicly available, but are available from the corresponding author on reasonable request.

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
