# Peer review of "The Effects of Ying Yang Bao on Nutritional Status of Children Aged 6–60 Months in Underdeveloped Rural Areas of China"

_nutrients, 2024, doi:10.3390/nu16020202_

Round 1
Reviewer 1 Report
Comments and Suggestions for Authors
I am very grateful you for the invitation to review manuscript nutrients-2723418 by Feng and coauthors "Ying Yang Bao Improved the Nutritional Status of Children Aged 6–60 Months in Underdeveloped Rural Areas of China”. The aim of this paper was to investigate the nutritional status of children aged 6–60 months in underdeveloped rural areas of China, and to evaluate the effectiveness of the 18-month Ying Yang Bao (YYB) intervention in improving the nutritional status of children aged 6–60 months. The work is interesting but needs adjustments to increase the quality of the material.
Comments:
- Abstract: Present the problem that resulted in the need to implement the study. The real motivation is unclear.
- Lines 32-33: Change the repeated keywords by different words from the title
- Introduction: Present the total number of children affected by the problem.
- Introduction: Better specify the composition of YYB.
- Line 91: Change “in FIGURE 1.” to “Figure 1”.
-Line 110: Specify the title of Figure 1 better.
- Line 142: Change “TABLE 1” to “Table 1”.
- Line 163: Change “TABLE 2” to “Table 2”.
- Throughout the entire text, replace FIGURE and TABLE with Figure and Table, respectively​.
- Discussion: Despite the discussion presented, authors must include information about anemia, people affected, associated problems, etc.
- Discussion: The nutritional aspects of YYB must be presented, as well as the role of each nutrient in the analyzed variables.
- Discussion: the discussion broadly presents other related works but does not actually discuss important themes to this work. Authors should deepen the discussion on the points of this work.
- Conclusion: It is superficial and needs to be presented more clearly, relating the points observed in the study.
Author Response
Dear reviewer,
Thank you for your letter and comments concerning our manuscript entitled “The Effects of Ying Yang Bao on Nutritional Status of Children Aged 6–60 Months in Underdeveloped Rural Areas of China” (nutrients-2723418). The comments are all valuable and very helpful for revising and improving our paper. Based on your suggestions, we have accordingly revised our manuscript. Attached you could find the point-to-point responses to the questions regarding the manuscript.
We hope that our answers have satisfied your comments.
Thank you and best regards.

Reviewer 2 Report
Comments and Suggestions for Authors
In the manuscript submitted to me for review entitled "Ying Yang Bao Improved the Nutritional Status of Children Aged 6–60 Months in Underdeveloped Rural Areas of China“ the authors Jing Feng, Yongjun Wang, Tingting Liu, Junsheng Huo, Qin Zhuo and Zhaolong Gong evaluated the effectiveness of implementing the 18-month Ying Yang Bao (YYB) in improving the nutritional status of children aged 6–60 months in underdeveloped rural areas of China.
I was very impressed with the research done, especially with the real applied meaning in the lives of a certain group of people for whom it would be vitally important. Everyone knows that there are areas of the world where people suffer from malnutrition and this has the biggest impact on children and pregnant women. But touching on a study like this that provides at least a temporary, partial solution raises optimistic thoughts that perhaps childhood malnutrition and the health problems that accompany it can be managed in the future.
The research was conducted over a period of four years from 2018-2021. The introduction to the manuscript presents the problem well enough. The methods used are well described. The obtained results are presented with the help of two figures and three tables and fully correspond to the conclusions made by the authors.
To support their research, the authors used 51 references presenting information on the matter from the last 20 years. Of the total number of references, 32 are from the last 5 years (nearly 2/3), which shows that the topic is current and deserves the reader's attention. I did not notice any redundant self-citations, all references used are relevant to the preparation of the manuscript.
My remarks and recommendations to the authors are:
1. In my opinion, Figure 1 should be renamed. The current title of the Flow Chat figure is not sufficiently descriptive of what is represented in it. The figure itself is very clearly presented - it gives a great overview of the structure of the study.
2. In section 3.3. and Table 3 gives the results of Model 1 and 2. I did not see the two models mentioned in the Materials and Methods section. I think it would be good to mention them there, even if only briefly.
3. I have a question that was born out of my personal curiosity and does not directly affect the research presented, but I would be interested if the authors would answer me. Is the receipt of YYB a policy of the state, of individual municipalities, or is it funded by a scientific project over a period of time to establish the benefits?
Author Response

(The authors gave the same response as above.)

Reviewer 3 Report
Comments and Suggestions for Authors
Dear Authors:
Regarding the manuscript with title “Ying Yang Bao Improved the Nutritional Status of Children Aged 6–60 Months in Underdeveloped Rural Areas of China”, I have some minor comments to address.
Minor Comments:
Comment 1: According to the Instructions for Authors of the Journal “Nutrients”, authors must: 1.1. delete the headings of Abstract
1.2. before presenting the purpose of the study, authors have to place the question addressed in a broad context.
1.3. The abstract should be a total of about 200 words maximum.
Comment 2: I suggest authors to change the title from ““Ying Yang Bao Improved the Nutritional Status of Children Aged 6–60 Months in Underdeveloped Rural Areas of China” to “The effects of Ying Yang Bao on Nutritional Status of Children Aged 6–60 Months in Underdeveloped Rural Areas of China”
Comment 3: Lines 46-47: I Kindly ask authors to specify what contains the free bago f nutritional pack?
Comment 4: Line 49: Authors must change “from 6 to 24 months old” by “for infants and young children aged 6-24 months”
Comment 5: Lines 57-59: “The LHEAPITNP focuses on nutritional status of children including hemoglobin levels (Hb) and anemia, physical development, and behavioral development.” Authors have to correct the gramar of the previous sentence.
Comment 6: Line 66: According to the Discussion and Conclusions presented on the manuscript, I suggest authors to delete the expression “the nutritional status of children, and”
Comment 7: On Statistical Analysis, authors must add information regarding the tests used to evaluate the differences between intervention and control groups regarding Basic characteristics of subjects.
Author Response

(The authors gave the same response as above.)

Round 2
Reviewer 1 Report
Comments and Suggestions for Authors
After carefully checking the responses and the revisions, the manuscript is suitable for Nutrients.
Author Response
Dear reviewer,
Thank you for your time and efforts in reviewing our manuscript entitled “The Effects of Ying Yang Bao on Nutritional Status of Children Aged 6–60 Months in Underdeveloped Rural Areas of China” (nutrients-2723418). It’s very glad to have your positive evaluation of our work. In the future studies, we will maintain our focus on the diet and nutritional status of children, and strive to contribute to the healthy growth and development of children.
Thank you and best regards.